# Nut Geometry Inspection Using Improved Hough Line and Circle Methods

**DOI:** 10.3390/s23083961

**Published:** 2023-04-13

**Authors:** En-Yu Lin, Ching-Ting Tu, Jenn-Jier James Lien

**Affiliations:** 1Department of Computer Science and Information Engineering, National Cheng Kung University, Tainan 701, Taiwan; 2Department of Applied Mathematics, National Chung Hsing University, Taichung 402, Taiwan

**Keywords:** nuts, computer vision, parallel, opposite side length, straightness, eccentricity, diameter, roundness, concentricity

## Abstract

Nuts are the cornerstone of human industrial construction, especially A-grade nuts that can only be used in power plants, precision instruments, aircraft, and rockets. However, the traditional nuts inspection method is to manually operate the measuring instrument for conducting an inspection, so the quality of the A-grade nut cannot be guaranteed. In this work, a machine vision-based inspection system was proposed, which performs a real-time geometric inspection of the nuts before and after tapping on the production line. In order to automatically screen out A-Grade nuts on the production line, there are 7 inspections within this proposed nut inspection system. The measurements of parallel, opposite side length, straightness, radius, roundness, concentricity, and eccentricity were proposed. To shorten the overall detection time regarding nut production, the program needed to be accurate and uncomplicated. By modifying the Hough line and Hough circle, the algorithm became faster and more suitable for nut detection. The optimized Hough line and Hough circle can be used for all measures in the testing process.

## 1. Introduction

Currently, industry 4.0 of smart manufacturing combines machines from different places, and the data can be sent by various kinds of sensors that remotely monitor product quality, quickly screening out defective products and reducing labor costs. Machine vision is widely used to eliminate product defect rates and improve the defect-free ratio, geometric measurement of objects, identification of different objects, control of robotic arm grasping, geometry inspection of defective items, and reduction of labor costs. With practical applications for the production line, defect items can be removed in advance once they are detected. Therefore, the number of defective products entering the market and becoming components can be reduced. This early disuse is especially important for products that focus on precision and need to be durable to support an object, such as nuts. A-grade nuts are the foundation of various fields of industry. Many products require nuts of good quality. The application of nuts can be as small as toys or as large as building a space station.

Nuts are an important cornerstone of human construction and production. Now, building a car needs 700 nuts, and building a Boeing 747 requires 3 million nuts. There are countless power stations and space programs. These practical applications all require nuts that fully meet the specifications. If the specifications of the nuts can be determined quickly to meet the standard of A-grade and be applied to a large quantity, the manufacturer could gain a better product price and net profit.

At present, there are two types of geometry inspection for nut manufacturers: manual and machine vision. Manual inspection is a random inspection and the inspection speed cannot keep up with the production speed. Concerning machine vision detection, the captured images need pre-processing before the inspection. Therefore, the Adaptive Gray Level Mapping algorithm (AGLM) [1] was proposed to improve image quality. However, AGLM only performs simple circle and line detection and cannot be used in real conditions [2]. For real conditions, a multi-feature hierarchical locating (MHL) algorithm was proposed. Furthermore, the Wiener filter [3] was applied to remove noise and restore the original image when an image is shaken and blurred. To remove image noise, a 3*3 Gaussian filter was used in this work to achieve the desired result.

Line detection was used to find the six sides of the nut. Then the information on these six sides helped to obtain the length of the opposite side. Finally, the six straight lines drawn were used to compare this with the real outline. A method combining Hough and Shi Tomasi was proposed [4], which improved Hough’s [5] traditional longer calculation time and the need for more memory. It was not possible to use the unimproved Hough line to find all of the sides of the nut, so the hexagonal nut angle had to be fixed to repair the missing edge [6]. Deep learning of the machine was applied to identify the nut in the picture [7,8,9], and Hough line detection was used to find the sides. The deep learning model of the Hough line also helped to detect the edge of the nut [10]. Detecting the edge by using a faster-region convolutional neural network (faster-RCNN) improved the use of the traditional Hough line [11]. Moreover, You Only Look Once (YOLO) v3 was applied to deep learning to identify defective nuts [12]. Regarding straightness detection, straightness had to be detected based on vision so the peaks and troughs of the target area could be discovered [13]. Once the peaks and troughs of the target area were found, one of the line segments was cut into smaller pieces of line segments. Each line segment was thus calculated independently [14].

Circle detection was used to find the center and diameter of the bore hole of the nut. The center and diameter of the inner hole would be further used to measure the degree of roundness, concentricity, and eccentricity. Hough circle is often used for circle detection, but it has the problem of too much calculation time and false circles. To shorten the calculation speed and reduce false circles, different methods were proposed. First of all, reducing candidate points [15] by, using multi-segmented line segments [16,17], and using the line segments on the arc to detect the circle were suggested. Second, the concepts of the isosceles triangle [18,19,20] and the right triangle [21] were also proposed as potential methods to detect the circle on the arc. However, the methods could not provide a better solution for being too complex and having invalid voting. A third method suggested was angle-Aided Circle Detection [22]. Angle-Aided Circle Detection was proposed to solve the problem of computational complexity and invalid accumulation by dividing the image into sub-regions for sampling edge points. Another method for reducing the amount of calculation was image compression, which was suggested as a method to reduce candidate points [23]. Last but not the least, edge segmentation, which is one of the methods for classifying different types of edges [24], was proposed. With edge classification, the circle could be detected. However, this method can only be used in certain conditions. Each pixel in the image corresponds to the three-dimensional parameter spaces for weight voting. Using a parameter space proportional to the radius avoids eliminating large circles with the same threshold and removes false small circles [25]. Nevertheless, different thresholds needed to be set for different images. The Hough circle detection results were improved using the Vector Quantization algorithm [26].

Regarding visual inspections of roundness, the traditional method for roundness inspection [27,28,29] was to use a probe tester, and the roundness would be determined by connecting the probe and the measured object. Adding Zernike and sub-pixel could also optimize the edge detection results to optimize the canny detection before carrying out Hough roundness inspection [30]. These studies [31,32] provide interesting and innovative methods to address issues related to materials, science, and engineering. The results of these papers contribute important insights and understanding to the field of research and have inspired further research in related areas.

The Section 2 of this work introduces the testing specifications and equipment. The Section 3 introduces the implementation method. The Section 4 shows the experimental results. The Section 5 is the conclusion.

## 2. System Setup

The detection equipment in this thesis is divided into three parts. As shown in Figure 1, the camera is an industrial camera and a turntable with a rotation speed of up to 0.6 m/s is used to simulate a conveyor belt or turntable. Subsystem1 is a dynamic capture, and the second subsystem is nut geometric detection. For Subsystem1, the object would be automatically captured when it passes through the center of the screen, and then the real-time image would be displayed on Subsystem1, and the captured image would be sent to subsystem2 for geometric inspection.

Table 1 is our nut inspection specification. We used A-grade nuts with an error of only 1% as the detection standard. If the detection does not meet the detection standard for A-grade nuts, the detection system will identify that the nuts are not A-grade nuts. Currently, the precision-grade nut, also known as an A-class nut, is only defined in the international standard DIN 934, with three specifications: a length across flats of 13 mm, a thickness of 6.5 mm, and an inner diameter range of 6.91 mm to 6.65 mm. Therefore, the standard inner diameter is 6.78 mm, with a tolerance of ±2%. To inspect the nuts produced by the thread-cutting machine team, we adopted a more stringent tolerance of ±1%. We propose definitions for testing items that have not been defined by other international standards.

In subsystem2, we could find the six sides of the nut (L1~L6), the six corner points P1~P6, the center of the triangle T, the length of the opposite side $L_{i}$, straightness $S_{i}$, diameter D, the coordinates of the center of the circle C, roundness $R_{d}$, and the Concentricity $C_{t}$, eccentricity Ed, as shown in Figure 2. As shown in Figure 1, the simulated environment involved the nuts advancing on the conveyor belt at 0.6 m/s with a ring-shaped white light. When the nuts passed through the center of the screen, they would be automatically captured. The real-time image would be displayed on the upper left area of the screen, and the captured image would be displayed on the upper right area. A small screen displayed the first 8 captured images.

## 3. System Framework

The geometry inspection of this study includes automatic image acquisition and pre-processing, parallelism detection, opposite side length detection, flatness detection, diameter, roundness detection, concentricity detection, and eccentricity detection, as shown in Figure 3.

### 3.1. Dynamic Capture Subsystem

To automatically capture images for nut geometry inspection, the production line conducts the real-time inspection. The calculation speed of the entire inspection system is faster than the production speed of the nuts. Therefore, Moment was used to automatically capture the nuts that pass through the center of the screen. With this method, the coordinates with the highest pixel density in the image were calculated. When the highest density coordinates pass through the center of the screen, the acquisition system would automatically capture the image for the inspection system to work on further detections.

Circular area:(1)$$m_{00}=\sum_{x=1}^{M}\sum_{y=1}^{N}f(x,y)$$

The coordinates of the center of gravity coordinate:(2)$$m_{10}=\sum_{x=1}^{M}\sum_{y=1}^{N}x(x,y)\;m_{01}=\sum_{x=1}^{M}\sum_{y=1}^{N}y(x,y)$$

$m_{00}$ is the zero moment. $m_{10}$ and $m_{01}$ is the first moment. *M* and *N* are the sizes of the input arrays. ($\frac{m_{10}}{m_{00}},\frac{m_{01}}{m_{00}}$) = ($x_{0},y_{0})$ is the nut center. The function f(x,y) is normalized to be 0.0~1.0, which is used as a weight. Possible variables that reduce the accuracy of the image inspection include dust on the conveyor belt, which might be iron fillings generated by tapping. However, the dust and iron fillings would not cover the nut because a high-pressure lubricant was used to clean the surface. Another factor is the noise generated by the camera working long hours. Therefore, the captured image is processed by a Gaussian filter, which is a common way to remove the noise. In this case, a 3 × 3 Gaussian filter was used.

### 3.2. Inspection Subsystem

The purpose of conducting the line inspection is to obtain detailed parameters related to the edge of the nut. To achieve high precision and detailed information concerning edge detection, three items were examined: parallelism, opposite edge length, and flatness. The automatically acquired pre-processed image was used for the line inspection. The first objective was to obtain the six edges of the hexagonal nut using Sobel edge geometry inspection [33], which is used to calculate the gradient magnitude $G_{p}$ and the gradient direction $\theta_{p}$:(3)$$G_{p}=\sqrt{{G_{x}}^{2}+{G_{y}}^{2}}$$
(4)$$\theta_{p}=atan2(\frac{G_{y}}{G_{x}})$$

$G_{p}$ is the gradient magnitude. $\theta_{p}$ is the gradient’s direction. As shown in Figure 4, the gradient magnitude ranges from 0 to 255. To remove noise in $G_{p}$ and $\theta_{p}$, a gradient magnitude threshold of $t_{g}$ = 50 was set. Any point p in $G_{p}$ and $\theta_{p}$, was removed when $G_{p}<t_{g}.$ Next, a hexagonal nut has six edges, so there would be six peak values of gradient direction. The figure of Magnitude-direction was plotted as a histogram $I_{h}$, *x*-axis: the gradient direction was θ, and the range was −180 to +180 degrees, with 3 degrees per bin. The y-axis shows the number of points with a gradient direction of θ. We kept the six sides and removed non-edge pixels. Next, we set the maximum value of $I_{h}$ as $m_{\theta }$, and remove the number of points with gradient direction less than $\frac{1}{2}m_{\theta }$. Therefore, we could obtain the angle $p_{i}$ of the six edges of one nut.

To obtain the exact angles of the 6 sides, the original setting was 3 degrees per bin and then converted to one bin per degree. $a_{i}$ represents the angles of a hexagon’s six sides, denoted by i = 1~6. If $p_{i}$ represents the weight, the angles of $a_{i}$ would range from 57° to 63° degrees. $N_{i}$ represents the total count of pixels for each side. By calculating $a_{i}$, we can obtain the angle of 6 sides. $\sum_{j=0}^{6}\left(p_{i}+j-3\right)$ refers to the number of coordinate points in the image at that angle. $p_{i,j}$ represents the candidate angle:(5)$$a_{i}=\frac{\sum_{j=0}^{6}({p_{i}+j-3)p}_{i,j}}{N_{i}}.$$

As shown in Figure 5, the Hough line was conducted. All of the lines found were stored in the way of pole coordinates (r and θ), and the strongest line (the line that matched the most points found) was ranked as the largest with the Hough line. The start point and end point found along the line were recorded as one line. Since the angle found by the Hough line ranged from 0° to 180°, the gradient direction we conducted ranged from −180° to 180°. If the Hough line was 30°, the gradient direction would be 30° and 210° (=30 + 180). The key is to tell which one is 30° and which one is 210°.

As shown in Figure 5, to determine which line of each group of opposite sides is close to the origin nut passing the screen, the upper left corner of the screen was used as the starting point. The Hough line storage format is r and θ. The line with the closest θ and the smaller value of r is the line closer to the origin nut.

#### 3.2.1. Parallel Line Detection

Parallel line detection was diagonally connected to determine the parallelism of nuts. One of the three pairs of parallel lines was taken from the midpoint as a start point, then to the endpoint of each pair, and finally back to the midpoints. These three pairs formed three lines. Two findings were shown when the three lines intersected at one point. First, as shown in Figure 6, the area of the triangle should be measured if the three lines have no intersection point. If the triangle area exceeds the threshold, the parallelism (of the nut) is not good. If the area is a regular hexagon, the distance between the opposite corners and the opposite corners is equal. If the distance between the opposite corners is equal, the lines will intersect at the center of the nut. If they are not equal, a triangular area will be formed in the center of the nut. The larger the difference in distance, the larger the triangular area.

#### 3.2.2. The Opposite Side Length

To calculate the distance between parallel lines, 6 sets of start points and endpoints were applied. A total of 2 coordinate points were then connected to show a straight-line equation and the 6 straight-line equations. The assumption is that the length of the opposite side (LOS) might be closer to the standard length, which would indicate that it is not easy for the nuts to fall off from the automatic lock nut machine. With the 6 equations, the 6 coordinate points $c_{i}$ can be found. By drawing the 6 coordinate points, the line $s_{i}$ connecting the points can also be drawn:

If there are two lines $l_{1}:a_{1}x+b_{1}y=c_{1}$ and $l_{2}:a_{2}x+b_{2}y=c_{2}$
(6)$$(a)\;x=\left|\begin{array}{cc} c_{1} & b_{1}\\ c_{2} & b_{2} \end{array}\right|/\left|\begin{array}{cc} a_{1} & b_{1}\\ a_{2} & b_{2} \end{array}\right|,\;y=\left|\begin{array}{cc} a_{1} & c_{1}\\ a_{2} & c_{2} \end{array}\right|/\left|\begin{array}{cc} a_{1} & b_{1}\\ a_{2} & b_{2} \end{array}\right|$$

As shown in Figure 7, i = 1~3 in the line segment $s_{i}$; the coordinate points of the 1/4, 2/4, and 3/4 positions were found on the line segment $s_{i}$; and the vertical line segment $O_{i}$ can be found in the straight-line equation $a_{i}x+b_{i}y=c_{i}$. The slope of $\alpha_{A}$, which is the slope of the vertical line, is $\alpha_{B}$, which can be obtained by the equation of B: y = $\alpha_{B}x+b_{B}$. Since $\alpha_{B}$ is perpendicular to $\alpha_{A}$, the relationship is $\alpha_{B}=-1/\alpha_{A}$, $b_{B}=y-\alpha_{B}\times x$. A vertical line can be drawn on the coordinate points of 1/4, 2/4, and 3/4 on the line segment $s_{i}$. By calculating the average value $W_{i}$ of three perpendicular lines, the quality of a cross-flat width length can be considered good if $W_{i}$, i = 1…3 are close.

#### 3.2.3. Edge Flatness of the Nut

The fluctuation of the real edge on the calculated edge was calculated. We first used Sobel to extract the nut contour $I_{S}$ to avoid being affected by noise and non-edge points. Then the maximum threshold value $k_{max}$ = 2 of the distance from the point to the line was set. For every pixel *p* in $I_{S}$, which is distant from $s_{i}$ for less than $k_{max}$, the distance from *p* to $s_{i}$ was calculated. The pixel which had the largest distance to the $s_{i}$ on the left and right sides of $s_{i}$ was then calculated. Finally, farthest pixel on the left and right sides $f_{i}$ was added to complete the calculation. As shown in Figure 8, if $f_{i}$ is small, then the flatness of the nut is good.

#### 3.2.4. Bore Diameter

The diameter that determines whether the nut is easy to loosen or seize is shown in Figure 9. The Hough circle is used to detect the diameter of a circle. However, the Hough circle has the problem of taking too much time to detect the circle. To reduce the time of geometry inspection, fixed specifications were applied to reduce the Hough voting geometry inspection time. To obtain the required radius and center coordinates accurately, the edge detection result $I_{s}$ from Section 3.2.3 was used to draw a circle. Each pixel point on the circle was seen as the center point, with the bore radius r ± 5 pixels. Draw a circle around all pixel points in the image. The peak values of the 11 votes were compared, and the highest voting value r was used as the real radius. The top 3 peak values of this radius were then chosen to use a subpixel. Finally, a true circle center C ($x_{c}$, $y_{c}$) graph was drawn.
(7)$$\mathrm{General}\;\mathrm{form}:\;{\left(x-a\right)}^{2}+{\left(x-b\right)}^{2}=r^{2}$$
(8)Parametric form: x=a+rcosθ, y=b+rsinθ (In x-y domain).
(9)Parametric form: a=x+rcosθ, y=b+rsinθ (In a-b domain).

#### 3.2.5. Roundness

As shown in Figure 10, the roundness was calculated. If the roundness is not good enough, the nut might be unevenly stressed. The uneven force might cause damage and result in the nut falling off. The center coordinates taken from Section 3.2.1 and the radius r ± 15 pixel were used to draw two circles, and the ROI range is between the two circles. The distance D0~D359 from C(x,y) to each angle and edge of the bore was calculated, within which the $D_{max}$–$D_{min}$ were considered to indicate true roundness R_d_. When R_d_ exceeds the threshold value, we know that the bore is not qualified.

#### 3.2.6. Concentricity

Only when the eccentricity is known can the position of the fixture and the tapping be adjusted. Whether or not the coordinates of the center of the circle coincide with the coordinates of the center gravity of the nut reveals the distance between the center of gravity T of the nut and the center C, as shown in Figure 11. To obtain concentricity, the distance between the center of gravity T of the nut and the center C of the bore should be calculated.

#### 3.2.7. Eccentricity

To calculate the displacement of the center of the circle before and after tapping, whether tapping would shift the nut’s center of gravity was investigated. Throughout the process, the datum of the captured image had to be consistent. As shown in Figure 12, regarding eccentricity, the change in the central coordinates before and after tapping needs to be noted. However, the images before and after tapping were different. Therefore, it was necessary to collect the datum for calculating the eccentricity first, and then transform the image into a consistent form based on the datum.

The findings suggested that tapping would not change the shape of the nuts. However, the nut would not meet the standard of A-grade specification if the shape were to change. This feature can be applied to determine eccentricity. Since the tapped nut would not deform, the center of gravity of the nut and the length of the opposite side remained unchanged. The center of gravity of the nut was taken as the center of rotation, which translated the images before and after tapping and made the center of gravity the center of the screen. The images before and after tapping were then rotated to the same angle. Calculate the rotation angle of the centerline. After the comparison, the longest centerline before tapping was used as the datum for image rotation after tapping, and the longest center line after tapping was rotated to resemble the same slope as the longest center line before tapping.

## 4. Experimental Results

The geometry inspection method was tested and evaluated. The nuts were placed on the turntable/conveyor belt. Both normal nuts and abnormal nuts were mixed to simulate the actual production line. As shown in Table 2, we collected 50 normal nuts before tapping, 50 normal nuts after tapping, 24 abnormal nuts before tapping, and 20 abnormal nuts after tapping. The nuts were formed by press forging, and badly formed nuts were produced only when the production machine was faulty and the maintenance was negligent. The tapping machine team was requested to produce abnormal nuts for the test, using grinders to wear the edges and bearing surfaces and pliers to flatten the nuts. Nuts with three types of defects before tapping are shown in Figure 13, and nuts with three types of defects after tapping are shown in Figure 14. The defective nuts are divided into three types: line defects, round defects, and flat defects. In this study, the purpose of using nuts with three types of defects was to compare the results caused by different types of nut damage.

We are currently developing the system in a laboratory environment and have found the method to be feasible. To implement it on the production line, we still need to order some equipment, so we will need the company’s cooperation to proceed.

The nuts with line defects have unqualified edges, those with round defects indicate deformation of the bore, and bearing surface abnormalities refer to the grinding part of the bearing surface. The original image was 2048 × 2048 pixels and the central part of 700× 700 pixels was selected by a fixed distance and focal length. We resolved that 1 pixel was equivalent to 36 μm, and the nut width was 350 × 350 pixels. A range that was double the size was applied, which was 700 × 700 pixels. The results showed that 700×700 pixels were the most suitable size for capturing nuts moving at high speed. The nuts were placed on the 0.6 m/s turntable. When a nut passed through at high speed, the automatic capture system would capture its image. The proposed geometry inspection system is supposed to sensor from different directions and positions on the nuts. For regular nut machines, the nuts are fixed to a certain position and direction on the tray conveyor, and a backlight is attached to the transparent rotating disk. The geometry inspection machines are usually added after tapping. For traditional geometry inspection, requirements, 2 geometry inspection machines would be needed both before and after the tapping, which takes up more space and increases the cost. One of the advantages of the proposed geometry inspection method is that screening the image does not need additional machines.

In order to quickly obtain the information on the 6 sides of the nut, missing edges and false information had to be avoided. For example, in Figure 15, the Hough line probabilistically did show spurious edges, and the edges found were incomplete. This work proposes to optimize the Hough line so that the program will not be affected by light and shadow when detecting the 6 edges of the nut. The detected sides were divided into 3 groups, and three distances were taken for each group to calculate the length of the opposite side. The drawn side was then overlapped with the outline of the nut to calculate the straightness. Regarding circle diameter detection, false circles, which are not the target circle, may have been detected due to light and shadow. For example, Taubin may find a circle that is not the target, and the borehole information found may also be affected by light, shadow, and threads. Therefore, we improved the Hough circle and used weight to reduce the influence of the thread, as shown in Figure 16. We used the circle center coordinates to calculate the roundness. The concentricity is the distance between the circle center and the nut center, and the eccentricity is the change in the position of the circle center before and after tapping. We have improved the errors and false detections of the edge and bore when detecting the nut. To detect the nut faster than the nut manufacturing speed, the rest of the inspection only needs information on the nut edge and bore. We propose nut inspection items that do not require visual or contact instruments and hope that these inspection items can contribute to improving the quality of maintenance nuts.

The preset qualification of the proposed method is within ±1% of the standard. This method can be used to determine whether to replace the tap when the number of unacceptable nuts exceeds a certain limit. The before-tapping mean difference (BMD), before-tapping mean (BM), and before-tapping standard deviation (BSD) of the inspection results were calculated. The after-tapping mean difference (AMD), before-tapping mean (BM), and before-tapping standard deviation (BSD) of the inspection results were also calculated.

In Table 3, normal inspection nuts are shown. The three items for normal screw thread detection showed no changes before and after tapping, and in the normal nut circular inspection, the circular diameter difference showed to be merely 0.01 mm, indicating no significant difference. The results of roundness and concentricity after tapping were shown to be better than those before tapping, which is proof that tapping would not affect nut forming.

Regarding abnormal line detection, the manufacturing team of the tapping machine slightly polished the nut contour to simulate the contour of an abnormal nut. Regarding the geometry inspection of the bore, the unqualified bore was produced by the tapping machine manufacturing team, who made the bore out-of-round with pliers. Other methods were not used because the bore of the M8 nut is too small for tools to cause damage.

Damaging the nuts using rust or electrolysis to create an uneven edge of the worn nut was not considered either. This method could cause the whole nut to become larger or smaller evenly. Thus, the team in charge of operating the tapping machine decided not to produce abnormal nuts by using electroforming. In addition, no manufacturer would sell rusty nuts. Therefore, the use of electrolysis and rust to produce abnormal nuts was not implemented, either. The unthreaded nuts were ultimately provided by the manufacturer that collaborated with the tapping machine team.

### 4.1. Line Geometry Inspection Results

Table 4 shows a comparison of the results of the screw thread inspection between the normal and abnormal nuts. The inspection results for nuts with abnormal parallelism showed a 3–5 times higher standard deviation compared to the normal nuts. Moreover, the standard deviation for the length of opposite sides for abnormal nuts was 5–9 times higher than that of the normal nuts. Regarding the straightness detection, there was no significant difference before and after tapping because the edge of the nut was all flatly damaged by a grinder. Regarding eccentricity detection, the eccentricity of the normal nuts was shown to be better than the abnormal nuts. The findings suggested that tapping would not change the shape of the nuts, but only that of the bore.

### 4.2. Circle Geometry Inspection Results

As shown in Table 5, the standard deviation of the circular diameter decreased by 50% after thread cutting, while roundness and eccentricity remained unchanged. The results indicate that thread cutting can improve the bore diameter of the nut but cannot correct roundness and eccentricity abnormalities. The inability to justify roundness and eccentricity is due to the fact that not all internal threads can be accessed and cut when cutting threads on nuts with roundness abnormalities. Once a thread was not accessed and cut, the roundness and eccentricity could not be corrected or improved. Based on the results, it can be concluded that thread cutting cannot convert abnormal nuts to A-grade nuts.

## 5. Conclusions and Future Work

This work is about the manufacturing and geometry inspection of A-Grade nuts with a tapping machine, which has to be instant and accurate. The first advantage of the proposed inspection system is that it does not require the additional purchase of inspection machines. Second, the proposed system directly installs the camera to the existing tray conveyor, and at the same time, it does not need a backlight. This is an advantage because manufacturers usually use transparent grooves and backlights to remove noise from the image. Third, even if the results of the tapping are not satisfying, they could still be used to adjust the tapping machine.

The errors and incomplete edges in Hough line detection for the edges of a screw cap were reduced significantly with the proposed method, which is quick and accurate. The proposed method is a way to determine parallelism without angle measurement, instead using the shape of the screw cap. To better detect defective nuts, three equidistant vertical lines were used to weigh the length of each set of opposite sides. For straightness, we detected the straight line using Hough Line and found the overlapping area between the line and nut contour. Then, the maximum distance of the nut contour on both sides of the line in the overlapping area was calculated. Aoi was then applied to reduce noise and non-edge features. The position of the central gravity of the nut and the opposite side were used to calculate the offset of the center of the bore before and after tapping.

This study draws six conclusions: (1) Hough voting can be used for measuring the diameter of the circle, and the voting result can be used for weighting; (2) the fixed standard characteristic of the nut can be used to filter the radius search range and increase the calculation speed; (3) roundness is determined by whether the bore is round enough; (4) the radius center obtained by the weight can be used as a reference to scan the bore at 360°; (5) concentricity might show whether the central gravity of the nut and the bore overlap; and (6) new specifications are proposed for A-grade nut inspection, including parallelism, straightness, roundness, concentricity, and eccentricity. Due to the fact that the international DIN 934 precision specification only defines the length of opposite sides, thickness, and inner diameter, this study seeks to propose new specifications for the inspection of A-grade nuts.

This study provides a comparison regarding the calculation efficiency and quality of inspection between traditional deep-learning methods and the proposed machine-learning method. With the proposed method, accuracy and efficiency can still be maintained under complex conditions by merely detecting the six sides and the bore of the nut. For future research, innovative designs for a camera with higher accuracy are recommended, which might help to optimize inspection accuracy.

## Figures and Tables

**Figure 1 sensors-23-03961-f001:**
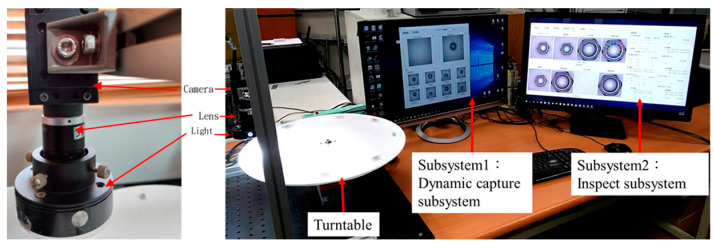
The inspection system is divided into three parts. Turntable simulation in the production line. Subsystem1 displays the status of dynamic capture. Subsystem2 displays the results of the geometric inspection.

**Figure 2 sensors-23-03961-f002:**
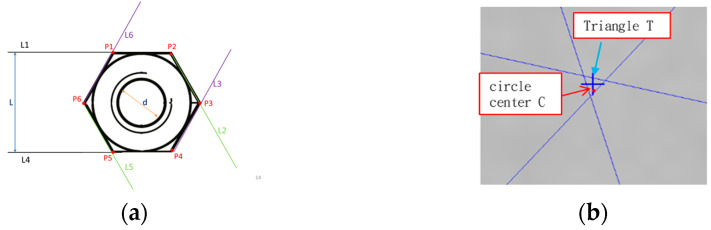
Defining the information: (**a**) code name; (**b**) center code name.

**Figure 3 sensors-23-03961-f003:**
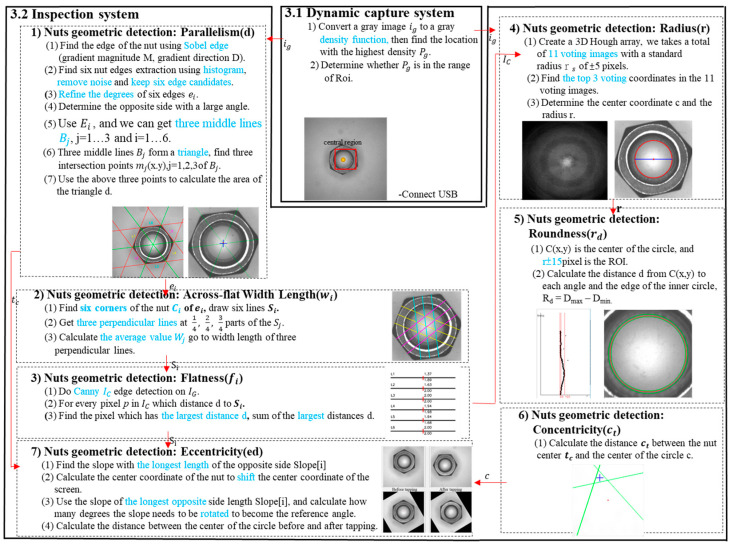
System framework.

**Figure 4 sensors-23-03961-f004:**
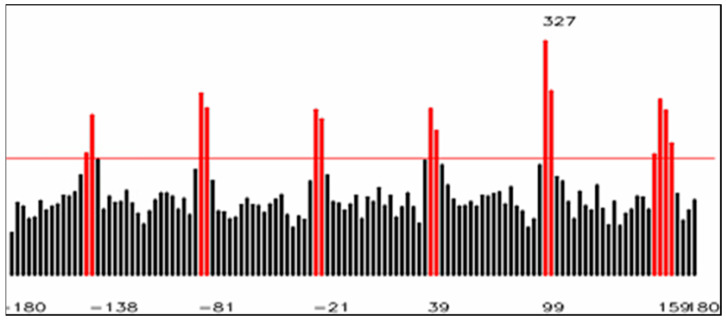
Using gradient direction to find out the six directions.

**Figure 5 sensors-23-03961-f005:**
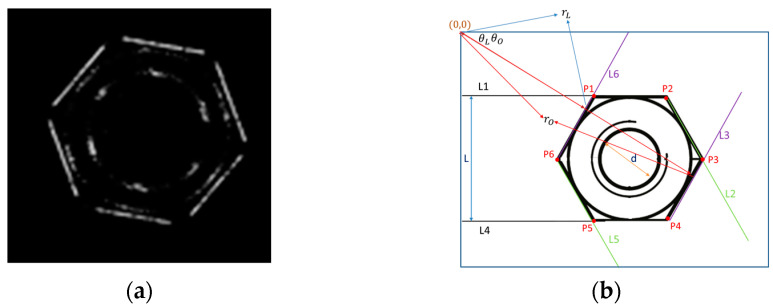
Determining the angle of each edge: (**a**) Displaying only the directions of the six edges; (**b**) Method of determining the angle of opposite edges.

**Figure 6 sensors-23-03961-f006:**
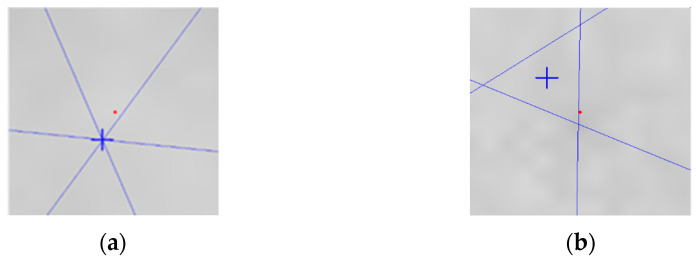
Using the intersection of diagonally adjacent lines to determine parallelism: (**a**) Intersected; (**b**) Not intersected.

**Figure 7 sensors-23-03961-f007:**
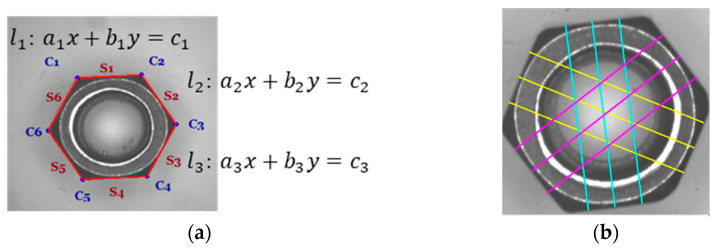
Calculate the length of the opposite side using three equally spaced perpendicular lines drawn for each parallel line: (**a**) Corner; (**b**) There are three sets of opposite side lengths, and each set has three lines.

**Figure 8 sensors-23-03961-f008:**
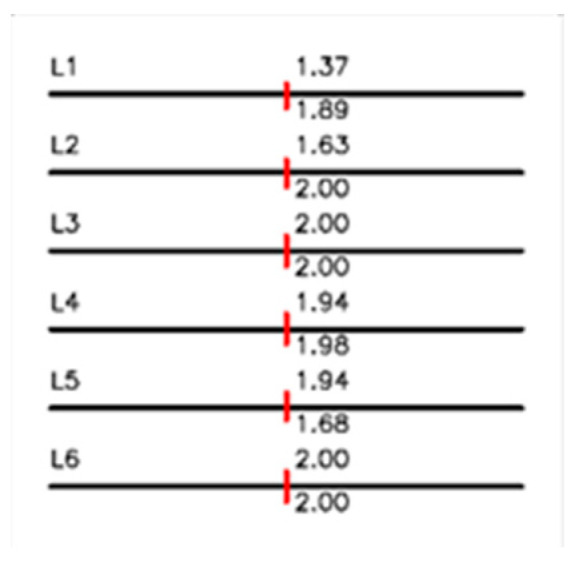
Calculate straightness by comparing the drawn lines with the contour of a nut. Display the farthest distance of the real edge from the fitted line segment on both sides.

**Figure 9 sensors-23-03961-f009:**
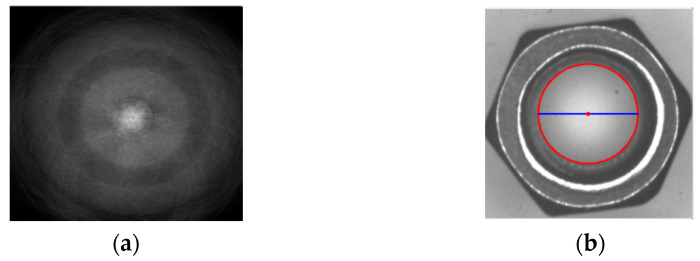
Using voting weight to find the center and radius of a circle: (**a**) Hough voting; (**b**). Radius.

**Figure 10 sensors-23-03961-f010:**
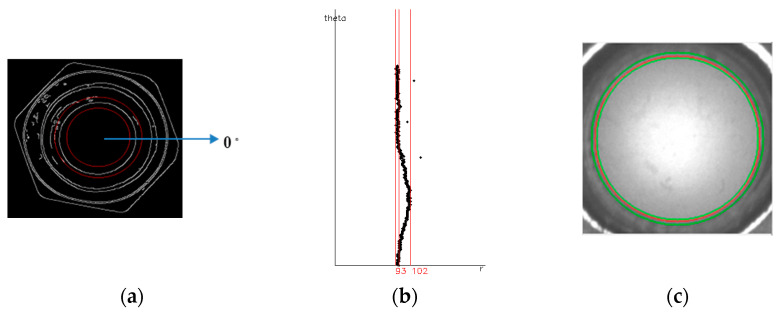
Calculating the distance between the angles of the center and the inner holes of a circle: (**a**) Search range; (**b**) D_0_~D_359,_ the red line represents the minimum distance from the center point, the average distance from the center point, and the maximum distance from the center point; (**c**). Roundness.

**Figure 11 sensors-23-03961-f011:**
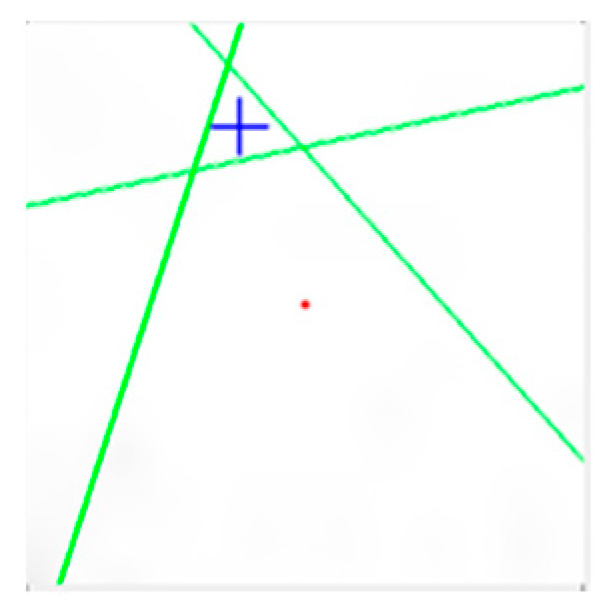
Using the distance between the center of the circle and the centroid of the nut to calculate concentricity. Blue indicates the center of gravity of the nut, green represents the connection of the midpoints of opposite sides, and red represents the center of the circle.

**Figure 12 sensors-23-03961-f012:**
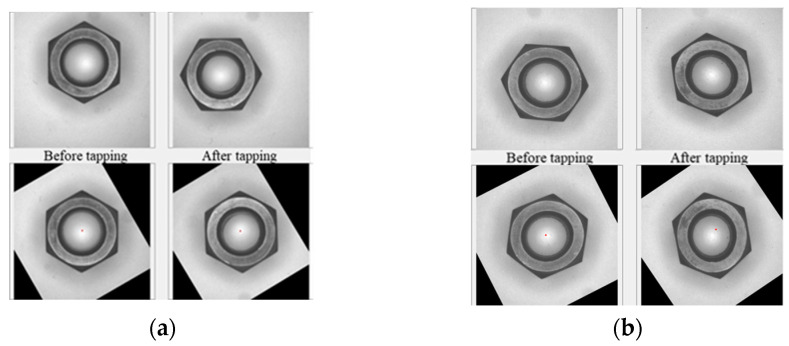
Using the coincidence of the nut before and after tapping to calculate eccentricity without damaging the outer appearance of the nut: (**a**) normal result; (**b**) abnormal result.

**Figure 13 sensors-23-03961-f013:**
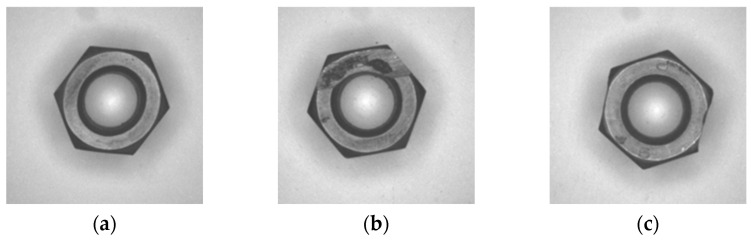
Abnormal nut before tapping: (**a**) circle anomaly; (**b**) plane anomaly; (**c**) line anomaly.

**Figure 14 sensors-23-03961-f014:**
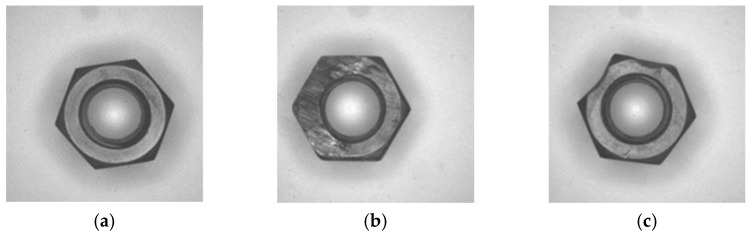
Abnormal nut after tapping: (**a**) circle anomaly; (**b**) plane anomaly; (**c**) line anomaly.

**Figure 15 sensors-23-03961-f015:**
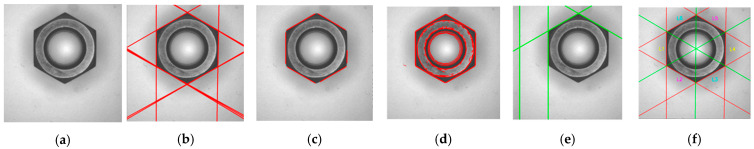
Comparison of other methods, the colored lines are the edges found by this method:(**a**) original; (**b**) standard Hough line; (**c**) Hough line probabilistic; (**d**) line segment detect; (**e**) Hough line use threshold; (**f**) ours.

**Figure 16 sensors-23-03961-f016:**
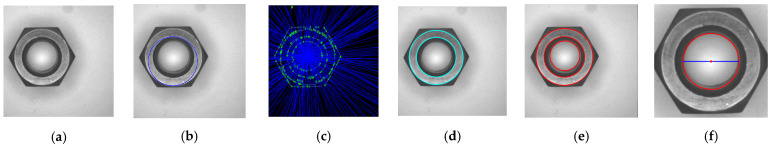
Comparison of other methods, the colored lines are the circle found by this method:(**a**) original; (**b**) standard Hough circle; (**c**) random circle detection; (**d**) Ransac circle; (**e**) Taubin; (**f**) ours.

**Table 1 sensors-23-03961-t001:** Nut specification.

Line Detection	Circle Detection	Line + Circle Detection
(1) Parallel 180° ± 1%	(4) Dimeter 6.78 mm ± 1%	(7) Eccentricity <12.86 mm × 1%
(2) Straightness <12.86 mm × 1%	(5) Roundness 6.78 mm × 1%	
(3) Opposite side length 12.86 mm ± 1%	(6) Concentricity 6.78 mm × 1%	

**Table 2 sensors-23-03961-t002:** Data collection.

	Normal	Abnormal
Before tapping	50	24
After tapping	50	20

**Table 3 sensors-23-03961-t003:** Normal nut detection.

	Parallel	LOS	Flatness	Diameter	Roundness	Concentricity	Eccentricity
BMD	$1.06^{{}^{\circ}}$	0.05 mm	0.01 mm	0.05 mm	0.05 mm	0.06 mm	
BM	$177.70^{{}^{\circ}}$	12.81 mm	0.12 mm	6.86 mm	0.29 mm	0.17 mm	
BSD	$1.15^{{}^{\circ}}$	0.06 mm	0.02 mm	0.60 mm	0.07 mm	0.07 mm	
AMD	$0.94^{{}^{\circ}}$	0.06 mm	0.01 mm	0.05 mm	0.04 mm	0.05 mm	0.03 mm
AM	$179.03^{{}^{\circ}}$	12.76 mm	0.13 mm	6.87 mm	0.22 mm	0.14 mm	0.16 mm
ASD	$1.05^{{}^{\circ}}$	0.08 mm	0.01 mm	0.07 mm	0.05 mm	0.06 mm	0.04 mm

**Table 4 sensors-23-03961-t004:** Line abnormal nut detection.

	Parallel	LOS	Flatness	Diameter	Roundness	Concentricity	Eccentricity
BMD	$2.33^{{}^{\circ}}$	0.50 mm	0.01 mm	0.02 mm	0.09 mm	0.14 mm	
BM	$178.88^{{}^{\circ}}$	12.48 mm	0.12 mm	6.83 mm	0.28 mm	0.23 mm	
BSD	$3.30^{{}^{\circ}}$	0.74 mm	0.02 mm	0.02 mm	0.12 mm	0.20 mm	
AMD	$4.01^{{}^{\circ}}$	0.28 mm	0.01 mm	0.02 mm	0.18 mm	0.13 mm	0.21 mm
AM	$178.95^{{}^{\circ}}$	12.60 mm	0.13 mm	6.80 mm	0.29 mm	0.30 mm	0.29 mm
ASD	$5.17^{{}^{\circ}}$	0.39 mm	0.02 mm	0.03 mm	0.21 mm	0.17 mm	0.24 mm

**Table 5 sensors-23-03961-t005:** Circle abnormal nut detection.

	Parallel	LOS	Flatness	Diameter	Roundness	Concentricity	Eccentricity
BMD	$1.57^{{}^{\circ}}$	0.13 mm	0.01 mm	0.08 mm	0.21 mm	0.06 mm	
BM	$179.19^{{}^{\circ}}$	12.75 mm	0.13 mm	6.89 mm	0.42 mm	0.18 mm	
BSD	$1.93^{{}^{\circ}}$	0.19 mm	0.01 mm	0.13 mm	0.26 mm	0.08 mm	
AMD	$2.87^{{}^{\circ}}$	0.11 mm	0.01 mm	0.05 mm	0.24 mm	0.06 mm	0.91 mm
AM	$179.185^{{}^{\circ}}$	12.80 mm	0.13 mm	6.84 mm	0.45 mm	0.19 mm	0.90 mm
ASD	$1.86^{{}^{\circ}}$	0.17 mm	0.01 mm	0.06 mm	0.27 mm	0.08 mm	1.09 mm

## Data Availability

Data is unavailable due to the private property rights of QST International Corp.

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
