# Peer review of "Nut Geometry Inspection Using Improved Hough Line and Circle Methods"

_sensors, 2023, doi:10.3390/s23083961_

Round 1

Reviewer 1 Report

Before the Editor makes a decision, I suggest that the authors must take into account the following corrections:

1.     The "Introduction" section should be more concise.

2.     It is not clear how were obtained the data in Table 1.

3.     Author must argue how the equations (3) and (4) were obtained, or indicate a bibliographic reference for their original form.

4.     Details on obtaining relation (5) are required.

5.     From where were taken the data used in Tables 3-8?

6.     Some editing "glitches" need to be corrected.

7.     Punctuations are used randomly. Insert comma or full stop after each and every equation accordingly.

8.     I think the authors need to emphasize more clearly the contribution of the manuscript from a scientific point of view.

9.     References are not uniformly written. See, for instance, [8], [12], [21].

10. Also, I think, the author must strengthen the References section with some articles that use some similar techniques, to make the techniques used more plausible, for instance: New analytical method based on dynamic response of planar mechanical elastic systems, Boundary Value Problems, Vol. 2020, No.1, Art. No. 104, 2020; An eigenvalues approach for a two-dimensional porous medium based upon weak, normal and strong thermal conductivities, Symmetry, 12(5), Art. No. 848, 2020.

If the authors take into account all these corrections, then this manuscript deserves to be published.  

Author Response

Response to Reviewer 1 Comments

Point 1: The "Introduction" section should be more concise.

Response 1: Move the portion of the original paragraph 99-120 to the experimental results; pp. 12.

Point 2: It is not clear how were obtained the data in Table 1.

Response 2: DIN 934 with three specifications: across flats length of 13mm, thickness of 6.5mm, and inner diameter range of 6.91mm to 6.65mm. So our standard is an inner diameter of 6.78mm with a tolerance of ±2%. We adopt a more stringent tolerance of ±1% to inspect the nuts produced by the thread-cutting machine team. We propose definitions for testing items that have not been defined by other international standards; pp. 3.

Point 3: Author must argue how the equations (3) and (4) were obtained, or indicate a bibliographic reference for their original form.

Response 3: Added to the references 31; pp. 17.

Point 4: Details on obtaining relation (5) are required.

Response 4:  represents the angles of a hexagon’s six sides, denoted by i=1~6. If  represents the weight, the angles of  would range from 57° to 63° degrees.  represents the total count of pixels for each side. By calculating, we can obtain the angle of 6 sides.  refers to the number of coordinate points in the image at that angle.  represents the candidate angle; pp. 6.

Point 5: From where were taken the data used in Tables 3-8.

Response 5: The screw was threaded by the tapping team and the results were obtained using the nut detection system built in this article, reorganize and explain the table 3-5; pp. 13.

Point 6: Some editing "glitches" need to be corrected.

Response 6: The content has been modified based on suggestions from other reviewers.

Point 7: Punctuations are used randomly. Insert comma or full stop after each and every equation accordingly.

Response 7: Thank you for the reminder. Punctuation has been inserted.

Point 8: I think the authors need to emphasize more clearly the contribution of the manuscript from a scientific point of view.

Response 8: The international standard DIN934 only defines the length、thickness、and inner hole diameter of the opposite sides. Therefore, we propose previously unaddressed inspection items, including straightness、parallelism、roundness、concentricity、and eccentricity. We hope that these inspection items can contribute to the specification of precision grade A nuts; pp. 15.

Point 9: References are not uniformly written. See, for instance, [8], [12], [21].

Response 9: Thank you for the reminder, it has been modified.

Point 10: Also, I think, the author must strengthen the References section with some articles that use some similar techniques, to make the techniques used more plausible, for instance: New analytical method based on dynamic response of planar mechanical elastic systems, Boundary Value Problems, Vol. 2020, No.1, Art. No. 104, 2020; An eigenvalues approach for a two-dimensional porous medium based upon weak, normal and strong thermal conductivities, Symmetry, 12(5), Art. No. 848, 2020.

Response 10: These studies provide interesting and innovative methods to address issues related to materials science and engineering. The results of these papers contribute important insights and understanding to the field of research and have inspired further research in related areas; pp. 17.

Reviewer 2 Report

Results can be improved with more precisions

Author Response

Point 1: Results can be improved with more precisions.

Response 1: We have modified the table of experimental results to make it easier to read, as well as updated the descriptions of the experimental results; pp. 12-15.

Reviewer 3 Report

1. Paragraph lines 99-120 show your proposed methods and benchmarks. It seems like it should be presented in the section “implementation method” or “experimental results”.

2. Please pay more attention to the format of your article.

2.1 For example, the “figure” in line 103 should be abbreviation as “Fig”.

2.2 Only vector and matrix can be bold.

3. Please explain all the symbols in your article.

4. Why 700*700 pixels is the most suitable size? I can’t find the result to support your conclusion.  

5. Please explain why the simulated scene can replace the actual production line. Is your method still applicable to the actual production line? Please prove it.

6. Please add the results of other benchmarks to show the robustness of your methods.

Author Response

Point 1: Paragraph lines 99-120 show your proposed methods and benchmarks. It seems like it should be presented in the section “implementation method” or “experimental results”.

Response 1: Paragraphs 99-120 have been moved to the experimental results, and the format has been modified based on suggestions from other reviewers; pp. 12.

Please pay more attention to the format of your article.

Point 2.1: For example, the “figure” in line 103 should be abbreviation as “Fig”.

Response 2.1: Thank you for letting us know that it has been modified.

Point 2.2: Only vector and matrix can be bold.

Response 2.2: The bold font used in the definition symbols has been changed to regular font.

Point 3: Please explain all the symbols in your article.

Response 3: The missing symbol definitions have been supplemented.

Point 4: Why 700*700 pixels is the most suitable size? I can’t find the result to support your conclusion.

Response 4: Our resolution is that 1 pixel is equivalent to 36μm, based on the calculation that the nut width is 350 * 350 pixels, we captured a range of double the size, 700 * 700 pixels; pp. 12.

Point 5: Please explain why the simulated scene can replace the actual production line. Is your method still applicable to the actual production line? Please prove it.

Response 5: We are currently developing the system in a laboratory environment and have found the method to be feasible. To implement it on the production line, we still need to order some equipment, so we will need the company's cooperation to proceed; pp. 12.

Point 6: Please add the results of other benchmarks to show the robustness of your methods.

Response 6: Currently, we haven't seen any other systems that analyze the geometry of the nut in as much detail as we do. However, since the international standard only defines the length across flats and the bore hole diameter, we cannot make a direct comparison with other systems; pp. 13.

Reviewer 4 Report

The authors proposed a novel nut inspection algorithm. The work is of interest and  the presentation is easy to follow. I just have a very few concerns:

1) Please locate figures and their corresponding captain in the same page, i.e., Figure 2.

2) Please make Figure 5 more clear, i.e., make the lines thicker and make the figure larger.

3) Please consider to compare the proposed results with normal detection in the same table to make the presentation more clear.

Author Response

Point 1: Please locate figures and their corresponding captain in the same page, i.e., Figure 2.

Response 1: Image position has been modified. All images should be placed on the same page or the following page.

Point 2: Please make Figure 5 more clear, i.e., make the lines thicker and make the figure larger.

Response 2: It has been enlarged; pp. 4.

Point 3: Please consider to compare the proposed results with normal detection in the same table to make the presentation more clear.

Response 3: Table 3 shows the results of the normal nut inspection. Table 4 shows the results of testing the abnormal screws with line defects, while Table 5 shows the results of testing the abnormal screws with circular defects; pp. 13.

Round 2

Reviewer 1 Report

The authors considered all my proposed corrections, which led to an improved form of the manuscript. 

Reviewer 3 Report

Please pay more attention to the format of your article.